# Understanding Antimicrobial Resistance Using Genome-Scale Metabolic Modeling

**DOI:** 10.3390/antibiotics12050896

**Published:** 2023-05-11

**Authors:** Tania Alonso-Vásquez, Marco Fondi, Elena Perrin

**Affiliations:** Department of Biology, University of Florence, Via Madonna del Piano 6, Sesto F.no, 50019 Florence, Italy; tania.alonsovasquez@unifi.it (T.A.-V.); marco.fondi@unifi.it (M.F.)

**Keywords:** metabolic modeling, antimicrobial resistance, bacterial metabolism

## Abstract

The urgent necessity to fight antimicrobial resistance is universally recognized. In the search of new targets and strategies to face this global challenge, a promising approach resides in the study of the cellular response to antimicrobial exposure and on the impact of global cellular reprogramming on antimicrobial drugs’ efficacy. The metabolic state of microbial cells has been shown to undergo several antimicrobial-induced modifications and, at the same time, to be a good predictor of the outcome of an antimicrobial treatment. Metabolism is a promising reservoir of potential drug targets/adjuvants that has not been fully exploited to date. One of the main problems in unraveling the metabolic response of cells to the environment resides in the complexity of such metabolic networks. To solve this problem, modeling approaches have been developed, and they are progressively gaining in popularity due to the huge availability of genomic information and the ease at which a genome sequence can be converted into models to run basic phenotype predictions. Here, we review the use of computational modeling to study the relationship between microbial metabolism and antimicrobials and the recent advances in the application of genome-scale metabolic modeling to the study of microbial responses to antimicrobial exposure.

## 1. Introduction

Antimicrobial resistance (AMR), the ability of a microorganism to resist the action of one or more antimicrobial agents, is one of the major public health problems of this century [1]. Antimicrobial-resistant microbes are currently estimated to claim ~700,000 deaths per year, and this mortality rate is predicted to increase to 10 million per year by 2050 [1]. Consequently, antimicrobial resistance has been identified as one of the most important challenges to human health by several national and international bodies.

For bacteria, the current antimicrobials inhibit a narrow spectrum of cellular processes (e.g., DNA replication, transcription, protein synthesis, and cell wall biosynthesis) [2]. Their action is traditionally seen as a linear process in which the antimicrobial enters the cell, reaches and interacts with its target, and stops the growth or kills the bacteria [3]. Accordingly, it is generally assumed that antimicrobial resistance relies on a few specific microbial genes [4]. However, this description represents only the immediate effects of the antimicrobial. In many cases, what is not known and is still the subject of debate, is how antimicrobials actually kill bacterial cells [5]. Microbial metabolism is impacted by antimicrobials [4], and in recent years, the close link between bacterial metabolism, antimicrobial action, and antimicrobial resistance has increasingly emerged. Indeed, on the one hand, antimicrobials alter the metabolic phenotype (metabotype [6]) of bacterial cells, giving rise to an altered metabotype. On the other hand, the metabotype of bacteria (normal or altered) influences their susceptibility to antimicrobials (Figure 1) [2]. Consequently, the antimicrobial efficacy can be enhanced by altering the metabolic state of bacteria. Approaches that target the bacterial central metabolism for drug development or that use the control of nutrients’ availability as a mechanism for the selection of antimicrobial-tolerant strains (for a definition of the differences between tolerance and resistance, see below) are gaining more and more attention [2].

Overall, antimicrobials can affect bacterial metabolism in three ways. First, they have a direct effect on metabolism that affects their efficacy, which seems to be different depending on whether the antimicrobial is bactericidal or bacteriostatic (even though it appears that all antimicrobials are ultimately bactericidal, and the difference between the two types of antimicrobials is only the rates at which they kill bacteria) [2,5]. 

The action of bactericidal antimicrobials on their primary targets causes damage to other essential macromolecules (nucleic acids, proteins, and membrane lipids) within the cell, resulting in the induction of stress response pathways. Stress responses increase metabolic activity to meet the corresponding energy demands. This results in the production of toxic metabolic byproducts such as reactive species, which damage macromolecules, and leads to the induction of additional stress response pathways. Once again, the overall cellular metabolic load results increased. The alteration of the metabotype, therefore, creates a cyclical process that ends with cell death [7,8,9,10,11,12,13,14,15]. Wong et al. [16] recently suggested that the interactions of the toxic metabolic bioproducts with the membrane induce a loss of membrane integrity, which results in cytoplasmic condensation through the leakage of cytoplasmic contents, and then in cell death. Alternative cell death pathways may involve cellular damage to nucleic acids and proteins resulting from both the primary drug–target interaction and from the subsequent generation of reactive metabolic byproducts [16]. A similar process, involving reactive metabolic byproducts, seems to contribute to antimicrobial lethality, also under anaerobic conditions [17].

Bacteriostatic antimicrobials, instead, inhibit protein biosynthesis or transcription in certain contexts, leading to a decrease in metabolic activity and subsequent cell stasis, thus, again resulting in an altered metabotype [14,18]. As mentioned before, however, bacteriostatic antimicrobials can also kill bacteria, probably depending, among other things, on the number of ribosomes. Indeed, when the number of ribosomes is reduced, ribosome-targeting antimicrobials seem to become increasingly bactericidal, suggesting a kind of lethal protein synthesis threshold. However, this could not be the ultimate mechanism by which these ribosome-targeting ‘bacteriostatic’ antimicrobials kill bacteria [5]. Therefore, the altered metabotypes resulting from antimicrobial treatment contribute to its final outcomes. The second and the third ways in which antimicrobials can influence bacterial metabolism are indirect. First, antimicrobial treatments generally involve the acquisition of resistance, through mutations or horizontal gene transfer, this often come with a fitness cost for the bacterial cell (Figure 1) [19,20]. Consequently, the acquisition of compensatory mutations that counterbalance the decreased fitness is a crucial step for the success of resistant strains [21,22]. These mutations generally restore normal growth, preserving resistance, and their number and type varies with the organism and the particular environmental conditions under which compensation occurs, indicating that fitness costs are dependent on the habitat and on the metabolic adaptation required for colonizing such a habitat [23]. Regarding the third mode, it has recently been demonstrated that, in addition to the acquisition of the classical mechanisms of resistance (target modification, drug inactivation, and drug transport), antimicrobials can also induce mutations in metabolic genes [24]. These metabolic mutations can confer resistance and are prevalent in clinical pathogens, suggesting that metabolic adaptation may represent a mechanism of resistance that confers tolerance, but may also mitigate the downstream toxic aspects of antimicrobials (Figure 1) [24].

On the other hand, the metabolic state of the bacterial cells can affect the antimicrobials’ efficacy in various ways. It must be stated that, in addition to the classical molecular mechanisms of antimicrobial resistance, bacterial cells can counteract antimicrobials in several ways, each of which relies on a general metabolic downregulation. Indeed, the term resistance is usually used to “describe the inherited ability of microorganisms to grow at high concentrations of an antimicrobial, irrespective of the duration of the treatment, and is quantified by the minimum inhibitory concentration (MIC) of the particular antimicrobial” [25]. If this resistant phenotype is observed only in a subpopulation of cells, it is known as hetero-resistance [26]. The term tolerance, instead, is used to describe “the ability, whether inherited or not, of microorganisms to survive transient exposure to high concentrations of an antimicrobial without a change in the MIC, which is often achieved by slowing down an essential bacterial process” [25]. It has been demonstrated that tolerance often evolves during frequent and intermittent antimicrobial treatments, and that its emergence often promotes the development of resistance [27]. Finally, the term persistence is used when only a subpopulation of a clonal bacterial population is able to survive exposure to high concentrations of an antimicrobial, without any genetic mutations [25]. Indeed, if persisters are isolated and regrown in the presence of antimicrobials, they display the same pattern of susceptibility as the original population [28].

The metabolic state of bacterial cells (normal or altered metabotypes, Figure 1) is mainly involved in mechanisms of tolerance to the antimicrobials’ action. Indeed, most of the so-called “phenotypic resistance”, in which metabolism has a fundamental role, is mechanisms of tolerance rather than resistance. Indeed, this term refers to all the transient situations in which a bacterial population, susceptible to an antimicrobial, becomes resistant without any genetic change taking place; therefore, resulting as not inheritable [29]. For example, the growth rate is an important parameter that affects the susceptibility to antimicrobials of bacterial populations. Resting cells are less susceptible to antimicrobials than metabolically active cells, especially to the action of bactericidal antimicrobials [4]. However, a recent study showed that the metabotype of the cell, instead of growth, better correlates with antimicrobial lethality, suggesting that antimicrobials should also be able to kill non-growing bacteria if metabolism is active, and that the metabolic response, following the initial interaction of an antimicrobial with its target, influences the bacterial response to the antimicrobial action [30].

Consequently, in all the conditions of an altered metabotype, in which the metabolism is poor or not active, bacterial cells are less susceptible to antimicrobials’ action (Figure 1). For example, during a stringent response, the accumulation of (p)ppGpp determines a global switch on bacterial metabolism that also modulates their response to antimicrobials [31,32,33,34], while bacterial persisters (whose formation also involved (p)ppGpp), as mentioned above, are subpopulations of metabolically repressed cells, which survive antimicrobial treatment, although lacking genetically encoded antimicrobial resistance determinants [35,36,37,38]. Additionally, during swarming motility, bacterial cells undergo a metabolic shift which makes them less susceptible to antimicrobials, through a mechanism that is poorly understood but that might be related to changes in the cell envelope [39,40,41,42,43].

Another example of phenotypic resistance is growth in biofilms, microbial communities embedded in an extracellular polymeric matrix [44,45,46]. Bacterial cells grown in biofilm are less susceptible to antimicrobials through a combination of resistance and tolerance mechanisms. Indeed, bacterial cells in biofilm are more resistant to antimicrobials than planktonic cells due to higher levels of both spontaneous and stress-induced mutagenesis, but also of horizontal gene transfer [45]. In addition, biofilm cells are also more tolerant to antimicrobials than their planktonic counterparts through a combination of different known mechanisms, in which the metabotypes of the cells play a key role. Indeed, biofilms are characterized by a gradient of nutrients and oxygen which decrease passing from the outermost to the innermost layers [46]. Consequently, they are composed of subpopulations with different metabotypes: metabolically active populations are located on the oxygenated and richer in nutrients surface of the biofilm, while non-growing subpopulations reside in the central anoxic and low in nutrient zones [45,46]. This stratified bacterial physiology corresponds to stratified layers of susceptibility to antimicrobials, with internal cells generally more tolerant than external ones [45]. In addition, the metabolic active cells at the surface exhibit increased expression of antimicrobial resistance genes, while the metabolically inactive subpopulation exhibits reduced or negligible expression of the antimicrobial targets and a reduced antimicrobial uptake [45]. Moreover, the gradient of nutrients and oxygen also induces a progressive activation of stringent and SOS stress responses that impairs the efficacy of antimicrobials, contributing to the antimicrobial tolerance of biofilms [45]. Finally, the presence of the biofilm matrix slows antimicrobials’ penetration and also favors their degradation due to the presence in the matrix of antimicrobial-modifying enzymes [45].

However, besides tolerance, bacterial metabolism is also involved in antimicrobial resistance. For example, classical elements involved in intrinsic resistance to antimicrobials, such as chromosomally encoded β-lactamases or multi-drug efflux pumps, play an important role in bacterial physiology, and they are not just an adaptive response to the presence of antimicrobials [4]. Furthermore, mutations in genes involved in cellular metabolism can contribute to intrinsic resistance [47]. Since, as stated before, bacterial metabolism is part of the antimicrobial-induced cell death process, mutations in the genes involved in these pathways (both their impairment and their overproduction) can influence the bacterial susceptibility to antimicrobials [4]. Moreover, all the regulatory genes involved in the control of metabolism, global regulators [48], signal transduction pathways controlled by two-component systems [49], and regulatory RNA [50] can influence bacterial susceptibility to antimicrobials. Finally, the metabolic adaptation that bacteria undergo during the colonization of a new environment (for example, during the early stages of host infection) might select antimicrobial-resistant bacteria (even in the absence of selective pressure with antimicrobials), thus further highlighting the existence of a tight link between bacterial metabolism and their susceptibility to antimicrobials [4].

In contrast to, for example, single mutations on specific genes that lead to antimicrobial resistance phenotypes, a causal relationship and/or mechanistic understanding of metabolism-dependent resistance/tolerance is much harder to achieve. Essentially, this is due to the inherent complexity of metabolic networks in which thousands of elements (reactions and metabolites) give rise to an intricate set of interconnections. 

One of the most powerful tools to study cellular metabolism at the system level is represented by genome-scale metabolic models (GSMMs) [51]. The starting point to work with GSMMs is the reconstruction of the metabolic network of the organism of interest, starting from its genome annotation. The methods that can be implemented to reconstruct metabolic networks include KBase server, Model SEED, or CarveMe [52,53,54]. The resulting models are then validated using several databases (such as KEGG, PubMed, and BiGG, the more the better [55,56,57]), genomes from related microorganisms, and further manual curation using experimental data as potential benchmarks. Ultimately, GSMMs are formal representations of cellular metabolism that include genes, enzymes, reactions, and metabolites, describing the associated gene–protein reaction (GPR) rules [58]. This facilitates computation and prediction of phenotypes through techniques such as Flux Balance Analysis (FBA) [59]. Using the optimization of an objective function of interest (usually cellular growth), these models are capable of predicting the activity (rate) of each reaction in the model. Despite being far from flawless, these simulations permit speculation on the role of specific reactions/pathways in the simulated conditions. Indeed, one of the most promising applications of GSMMs is the possibility, by key interventions on the model itself, to include environmental conditions in the modeling framework, i.e., researchers can opportunely tune some model parameters to represent the actual (experimental) intracellular and extracellular contexts in their GSMMs. The two most valuable examples in this context are represented by: (i) the possibility to define the nutritional landscape of the model by setting the boundaries of uptake reactions to mimic experimental nutrients’ availability, and (ii) the use of transcriptomic (as well as other -omics) data to define the set of enzymes that the cell is actually expressing in vivo. Both these methodologies constrain the model and create context-specific representations of the bacterial metabolism, according to the conditions used to obtain such extra information (nutrients’ availability and gene/protein/metabolite abundance). Since the first model ever developed, for *Haemophilus influenzae* [60], several groups have continued with this task, constructing models for other bacteria. Originally, GSMM have been exploited in metabolic engineering of biotechnologically relevant strains, as the optimization of natural products encoded by biosynthetic gene clusters (BGCs) [61,62].

GSMMs, however, have been progressively applied to many other areas of research. Recently, several works have illustrated the tools for the development of metabolic networks of Gram-negative pathogens and the metabolism of priority pathogens reported by the World Health Organization, the possible drug targets for them (antimicrobial pharmacology), and the awareness of the spread of antimicrobial resistance pathogens [63,64,65]. Here, we rather focus on the study of the intertwined relationship between microbial metabolism and antimicrobial resistance (including the development of novel drug targets) [66], and through a survey of currently available case studies, we show how GSMMs can improve its understanding. 

## 2. Case Studies

We have performed an operational classification of available case studies on the use of GSMMs to study antimicrobial (AM) exposure/resistance into antimicrobials: non-informed vs. informed (AM-non-informed vs. AM-informed) works. In the first group, we include all those works that did not use experimental data obtained from cells exposed to antimicrobial compounds to constrain and refine the model predictions. Conversely, all the studies that used experimental data (mostly -omics data) obtained from cells exposed to the antimicrobial molecule(s) to constrain the GSMM of the corresponding microorganisms (Figure 2) will fall in the second group. This latter group of studies will include, for example, those that used transcriptomic data obtained from microbial cells exposed to the antimicrobial compound to “inform” the model on which reactions can be used to represent the antimicrobial-exposed metabolic network. In other words, this approach permits to “turn on/off” those reactions whose genes are/are not expressed or active during the exposure to the antimicrobials.

All the metabolic reconstructions cited in this work and that have been used in anti-microbial resistance studies are reported in Appendix A.

### 2.1. AM-Non-Informed Models

Studying the metabolism of bacteria with the aim to prevent or fight AMR can be carried out by performing analyses such as gene/reaction or metabolite essentiality (GE and ME, respectively) in order to identify possible points of metabolic weakness of the cell. This approach removes a gene/reaction (or a metabolite in case of ME analysis) from the metabolic network and then simulates growth with FBA. In other words, every time a gene, reaction, or metabolite is deleted, an FBA is performed using the biomass production as the objective function. In 2009, in one of the first works of this kind, the *Acinetobacter baumannii*’s AYE GSMM was reconstructed and used in combination with a filtering framework of essential metabolites (EMFilter) to predict the most effective drug targets [68]. The first part of the analysis predicted 211 essential metabolites and, on this pool, the EMFilter was applied. The EMFilter is an algorithm that follows four steps. First, it removes currency metabolites that participate in reactions shared by several organisms, such as ATP and NADH. In the *A. baumannii* AYE case study, 179 metabolites remained after this step. Second, it selects essential metabolites that are surrounded by at least two metabolite-consuming reactions, in order to enhance the drug impact (97 metabolites remained). Third, it prunes essential metabolites that participate in human metabolism as well (22 metabolites remained). Fourth, it removes essential metabolites that are being consumed by human homologous enzymes, in order to avoid the possible interaction between the drug and the human (rather than the microbial) drug target. After all the steps of the filtering pipeline, the enzymes involved in the consumption of nine essential metabolites were predicted as final drug-target candidates. As a confirmation, some of the enzymes identified by this approach have been previously reported as drug targets.

Analogously to the *A. baumannii* AYE case study, Jenior’s group [69] decided to use this methodology to explore the possible metabolic pathway targets of *Clostridioides difficile*. They reconstructed the GSMMs of two strains (630 and R20291), and after the GE analysis, they compared the prediction of essential genes for growth in silico and in vitro, obtaining 89.1% and 88.9% accuracies, respectively. This group identified the minimum subset of metabolites necessary for growth, and subsequently, the carbon sources needed. Since *C. difficile* utilizes phase variation to generate phenotypic (rough- or smooth-edge colony morphology) and metabolic heterogeneity to maximize its fitness and virulence expression, Jenior’s group aimed to characterize the differences between in vitro and in vivo conditions using context-specific metabolic reconstructions derived from expression data. They found differences in alanine transport and utilizations, as well as in glycolysis’ transporters and metabolic reactions during periods of increased virulence expression. With the gene essentiality analysis, they found numerous reactions that were essential only for the smooth phenotype (especially in the pentose phosphate pathway, PPP), and surprisingly, no unique essential reactions for the rough phenotype. Finally, by comparing the predicted core metabolic activity of high- and low-sporulation conditions, Jenior’s group described how both N-acetylneuraminate and cytidine are being consumed during infection and decrease sporulation. These findings could help to find solutions to AM use in order to mitigate *C. difficile* infections. 

López-López’s group [70] also performed gene essentiality analysis in order to study what could be the best way to develop drugs for ampicillin-resistant *H. influenzae*. They refined its previous GSMM, and after the GE analysis, they found 11 genes of the fatty acid biosynthesis pathway (FASII pathway), 6 genes for the phospholipid biosynthesis pathway, and 12 genes encoding enzymes involved in lipid A biosynthesis. Since the FASII pathway generates acyl carrier proteins (key components of the bacterial membrane), these authors proposed that genes related to this pathway could work as a drug target, specifically, beta-ketoacyl-ACP, which initiates fatty acid elongation cycles.

To link the (complex) carbon sources that one microbe might encounter in vivo and the overall metabolic reprogramming during the infection process, Payne et al. [71] studied the metabolic responses of *Pseudomonas aeruginosa* to different mucins. Mucins are polymeric, highly glycosylated proteins produced by airway epithelial cells and submucosal glands that play a key role in the microbe clearance and infection prevention. Even though these proteins are not considered as antimicrobial agents, it is clear that the mucin–drug interaction may have a remarkable impact on drug absorption since mucus is the first barrier that drugs must overcome to be adsorbed [72]. This impact could be at the metabolic level, i.e., by changing the metabolic state of bacteria. Payne’s group developed a high-quality GSMM for *P. aeruginosa* (iPau21) that has 40 genes, 24 metabolites, and 76 reactions more than the previous model, iPau1129. This new high-quality model was constrained using transcriptomic data from the literature of four different conditions: casamino acids (no mucin exposure), MUC5AC, MUC5B, and mucin-glycans, all of which are present in different body parts prone to *P. aeruginosa* infection. Since the experiments were carried out with strain PAO1, its genes in the transcriptomic dataset were mapped to PA14 orthologs, and then the data were integrated with the iPau21 model. The integration of the data was performed using RIPTiDe [73], a tool developed to integrate transcriptomic abundances using parsimony of overall fluxes to identify the most cost-effective usage of metabolism that best reflects the maximization of the biomass production, which was the objective function. These four conditions led to four context-specific metabolic models that had a similar growth rate (less than five percent change). Payne’s group compared the metabolic mechanisms of all four conditions using a non-metric multidimensional scaling (NMDS). It showed that, even when there was no significant difference in the growth rate, exposure to mucin MUC5B caused the largest shift in the metabolic pathways utilized for growth, with MUC5AC in second place. Specifically, the fumarate and propionate metabolism differed between the control and the MUC5B models. This suggested that such a metabolic shift might be involved in the resistance mechanism of *P. aeruginosa*. Thus, they proposed AMs that target proteins for fumarate and propionate metabolism in order to combat mucin-resistant bacteria. Dahal et al. [74] recently refined the *P. aeruginosa* model (iSD1509) and added a whole new pathway for ubiquinone-9 biosynthesis, required for anaerobic growth. They validated the model in its totality using experimental data from different works, including Dunphy’s [75]. In order to understand how the metabolism could change under AM stress, Dahal et al. [74] performed flux sampling analysis using the biomass flux as an objective function, but constrained to 90% of the biomass flux value obtained by the FBA simulation, to mirror the presence of AM stress. In other words, the new biomass flux was constrained to 0.9 × the biomass flux value from FBA. This method aims to obtain several flux sets from the space of solutions in order to obtain a more robust result for each flux sustained by each reaction, and they indeed confirmed previous results in that the TCA cycle was found to be significantly upregulated in the fumarate-supplemented medium compared to the glyoxylate-supplemented one. Hence, the model correctly differentiated between metabolites that increased drug lethality versus those that did not and offered mechanistic explanations for these responses. In conclusion, confirming Payne’s group’s results [71], supplementing the medium with fumarate caused a higher oxygen uptake rate due to a higher TCA cycle activity, leading to a higher drug efficiency. 

Just as Payne et al. addressed the environmental–drug interaction scenario, evaluating how mucins change the metabolic state of bacteria, Chung et al. [76] proposed a different, metabolism-aware way to treat the multidrug-resistant (MDR) *Klebsiella pneumoniae* using polymyxin B in low doses, plus exogenous metabolites as adjuvants, which have a lower risk of resistance development. Chung’s group reconstructed the GSMM of four different *K. pneumoniae* strains, one polymyxin-resistant, one polymyxin-susceptible, and two polymyxin-susceptible but MDR. They studied seven different metabolites that were previously identified as significant metabolites perturbed by the combination with polymyxin B, at the level of both gene expression and metabolism of the same pathway. In silico predictions using flux sampling (i.e., without imposing an arbitrary objective function) revealed that the presence of at least three of these exogenous metabolites (3-phosphoglycerate (3PG), ribose 5-phosphate (R5P), and uridine 5′-diphospho-N-acetylglucosamine (UACGAM)) enhanced the central metabolism (higher fluxes of glycolysis, PPP, and TCA cycle), increasing bacterial growth. Since the growth rate is one of the factors that determines the phenotype of susceptibility to antimicrobials, with a slow growth rate being associated with low antimicrobial activity, 3PG, R5P, and UACGAM were good candidates for a non-antimicrobial plus polymyxin treatment. Chung’s group confirmed these results with time-kill studies, where bacteria displayed a decrease in the growth rate when using polymyxin B plus 3PG, R5P, or UACGAM, but not for the rest of the metabolites. Ultimately, this study provided evidence of the possibility that an altered metabolism could change the antimicrobial efficacy, also promoting the use of state-of-the-art modeling approaches to fully comprehend molecular adaptations to antimicrobial exposure.

An additional methodology to look for potential drug targets in different organisms is the combination of flux simulation-based techniques (e.g., FBA) with protein structure-based ones, such as structure-based virtual screening (SBVS). The latter is one of the most promising in silico techniques for drug design due to its robustness, and it is capable of predicting the most likely interaction between two molecules to form a stable complex [77], e.g., the drug and its target. Cesur and colleagues [78] used this protocol to find potential drug targets in *K. pneumoniae* through the constraint-based analysis of its GSMM. They simulated three host microenvironments by setting specific bounds to each reaction flux and set the biomass production as the objective function. Combining GE and ME analyses on this GSMM, Cesur’s group performed a preliminary screening that led to the identification of possible enzymatic targets, for which they identified the cellular compartments/function and performed a “druggability” assessment, i.e., to estimate the affinity of the protein to bind drug-like chemical compounds. Out of this selection procedure, one enzyme (encoded by the *kdsA* gene) stood out. For this target, using the computer-aided drug design approach (SBVS), Cesur’s group obtained two possible compounds, one of which showed to be quote promising as it is a derivative of coumarin with low acute toxicity and already proven high antimicrobial activity. Nazarshodeh and colleagues [79] also used the SBVS method to identify drug effectiveness, but for *E. coli*. To this aim, they used a GSMM-PRO, i.e., a genome-scale metabolic model integrated with protein structures. Combining GE analysis (biomass production as the objective function) with a search for informative protein–ligand complexes, they obtained 70 essential genes, accounting for 92 protein–ligand complexes, which were proposed as drug targets. To perform SBV screening, the authors used the 3D structures of FDA-approved drugs and ranked them against each essential target (ideally more than one for potential polypharmacology cases). Finally, their pipeline suggested eight known antimicrobials as possible drugs to target the essential genes found, proposing a new therapeutic indication for these reported drugs (drug repurposing).

A possible advancement for this work is represented by the simulation of drug epistasis, as drug combinations can illuminate new possible therapeutic strategies that might be inaccessible to single-target drugs, possibly eliminating functional redundancies exhibited by metabolic networks. In 2016, Krueger’s group [80] studied the possibility of extending FBA modeling in order to simulate drug effects over multiple doses. The methodologies implemented by this group (FBA-res and FBA-div) optimize the biomass production and account for the possibility that when a drug inhibits a reaction, (i) accumulated mass will slow down the upstream reaction thermodynamically (FBA-res), or that (ii) upstream reactions cannot sense or adjust to an inhibited downstream reaction on the timescale of the drug inhibition, and the substrate is wasted, creating less biomass over time (FBA-div). While these two theoretical approaches led to similar results to knockout simulations for single-agent effects, they had very different predictions for combinations with the approach simulating a flux-diversion to a waste reservoir (FBA-div), predicting potent antimicrobial synergies targeting metabolism. Indeed, FBA-div was able to model the strong antimicrobial synergies that target serial enzymes within a pathway, being able to predict synergies between three or more drugs. 

### 2.2. AM-Informed Models

One of the most worrisome threats in clinical microbiology is the increasing frequency of colistin resistance in *A. baumannii* [81]. Besides the important quest of new strategies to tackle the issue, we also chose this case study because, in our opinion, the work performed in the last years to tackle the issue of AMR in *A. baumannii* nicely underlines two main features of genome-scale metabolic modeling: (i) the scalability and versatility of GSMMs to be used with different experimental (-omics) datasets (even from previous studies, an important feature in the context of data reuse), and (ii) the possibility to constantly update and (possibly) improve pre-existing models once new data are available. In 2017, to address new possible solutions for the treatment of infected patients and to overcome the appearance of resistant phenotypes, Presta and colleagues [82] reconstructed the model of AMR *A. baumannii* ATCC 19606 to perform a systems-level study of the antimicrobial response in this bacterium. Specifically, to propose new targets for AM drugs, they performed GE analysis. Since the exposure to the antimicrobial might alter the metabotype (Figure 1), the authors used MADE (Metabolic Adjustment by Differential Expression) [83] to integrate the transcriptomic data obtained during colistin exposure (after 15 and 60 min of exposure [84]) with *A. baumannii*’s GSMM, leading [84] to four context-specific metabolic models (treated and untreated conditions at 15 and 60 min). After an FBA optimization using biomass production as the objective function, they compared the flux distributions for each treated part and its untreated counterpart using the flux ratio and looked for the pathways these reactions belonged to. The changes were mainly observed at the level of sugar and nucleotide metabolism and fatty acid biosynthesis. Besides the analysis of general trends, the combination of transcriptomic data and GSMM might in principle lead to the identification of those genes that become essential under specific growth conditions (context-specific essential genes), in this case following the exposure to colistin. In other words, the proteins encoded by these genes might represent promising drug targets when coupled with colistin treatment. Analyzing the essential genes for each condition, they found 9 and 5 genes that are essential only after 15 and 60 min of colistin exposure, respectively, and removing the genes with human orthologues (in order to exclude them as drug targets), they finally identified five essential genes, whose proteins could represent specific drug targets. 

The model reconstructed in this work (iLP844) led to further studies of *A. baumannii*’s metabolic network. Indeed, it was taken as a reference for the reconstruction of an improved model of the same strain (ATCC19606) [85] and for an *A. baumannii* AYE strain model (iCN718) [86]. Both groups used iLP844 to compare and validate their own model, by analyzing the shared genes and its growth on different carbon sources. Additionally, Norsigian’s group [86] used it to check and confirm reaction reversibility, and to better describe the transport reactions present in both models. This refinement allowed to interrogate the model on the interplay of multiple metabolic pathways under colistin treatment in *A. baumannii* and to shed light on some key points to optimize polymyxin combination therapy.

Another example is that of Banerjee and Raghunathan [87], who studied the case of metabolic reprogramming when *Chromobacterium violaceum*, an opportunistic human pathogen, is exposed to antimicrobials. They reconstructed *C. violaceum*’s model (iDB858). After experimentally measuring several cell rates, they constrained different uptake values of the model (glucose, violacein secretion, molar growth rate, ATPM, and oxygen) to represent three different conditions: wildtype and exposure to chloramphenicol and to streptomycin, both together and independently. Using the biomass production as the objective function, performing FBA, FVA, and GE, they described a rewired central and redox metabolism in the presence of each antimicrobial; specifically, high levels of NAD recycling provided by pyruvate, suggesting a reprogramming of metabolism to compensate for the stress consequences.

Researchers can take advantage of the already developed GSMMs for the organism under investigation. In this case, the integration of -omics data for the reconstruction of context-specific models (representing different metabotypes) can be performed in a more or less straightforward fashion. This is the case of Dunphy’s group [75], who used iPau1129 *P. aeruginosa*’s GSMM [88] and lab-evolved AMR lineages of carbon source uptake data from a previous work [89] to contextualize the potential impact of gene deletion in antimicrobial-resistant mutations (by setting the upper and lower bounds of the model according to the experimental data). Dunphy and colleagues profiled the metabotypes of four AMR lineages of *P. aeruginosa*: a lab-evolved phenotype, and piperacillin- (PIP), tobramycin- (TOB), and ciprofloxacin (CIP)-resistant strains. Each strain had been grown on 190 unique carbon sources, but there were only 14 growth-supporting carbon sources common for all four lineages, revealing a strong metabolic reprogramming following the exposure to each of the tested compounds. Since in Yen’s work they found that the PIP-evolved lineage contained a large deletion of 343 genes, much higher than for the other lineages, Dunphy’s group decided to analyze the role of each gene deletion and its impact on the metabolic functions and cellular growth. They did so by performing a GE analysis in silico using the biomass production as the objective function and repeated it on M9 minimal media and 42 different carbon sources, independently. The next step was to look for the intersection between predicted essential genes and the mutated genes in the PIP-evolved strain in order to predict resistance-specific essential genes and to identify incorrect model predictions. In conclusion, although the iPau1129 GSMM did not include several AMR genes, it was possible for Dunphy’s group to identify potential mutations impacting growth phenotypes, such as the loss of L-leucine catabolism in PIP-evolved metabotypes. 

Not surprisingly, most of the research concerning the effect of the cellular metabolic state on antimicrobial resistance emergence and/or antimicrobial efficacy has been carried out on *Escherichia coli*. In 2017, Zampieri et al. [23] used the most recent (to date) *E. coli* metabolic reconstruction (iJO1366) [90] in order to understand the impact of metabolic changes in the development of antimicrobial resistance. They selected three antimicrobials with different modes of action: ampicillin, chloramphenicol, and norfloxacin, and then let four independent lineages of the wildtype evolve resistance in minimal medium, either with glucose or with acetate as the sole carbon source. Then, they performed metabolome-based predictions using *E. coli*’s GSMM, and systematically maximized or minimized fluxes for each reaction of the model, i.e., they iteratively changed the objective function to calculate the shadow prices (the estimation of its sensitivity to changes in the availability of all individual metabolites). Negative shadow prices mean limiting metabolites for the reaction. In this way, reactions with overrepresentation of altered limiting metabolites could be those with an active role in the evolution of resistance. They observed that most of the evolved metabolic characteristics were present in pathways not directly affected by the antimicrobial. For the chloramphenicol-glucose-evolved populations, the evolved metabolic characteristics involved sugar transport, oxygen uptake, and CoA formation, while for the ampicillin-glucose-resistant populations, they involved nucleotide metabolism, serine biosynthesis, and cell wall recycling. The authors found an important difference between the ampicillin-glucose-evolved and ampicillin-acetate-evolved populations, with the first one being eight times more sensitive to fosfomycin, and the latter more tolerant to it. Thus, apparently, metabolic adaptation to antimicrobial exposure strongly depends on the nutritional context.

Recently, several groups have proposed the combination of GSMM and machine learning (ML) in order to broaden the space of predictions. ML is a method that handles the automatic learning of machines without explicit programming (black box) and has been widely used in the field of bioinformatics, and recently, in systems biology as well [91]. For example, in 2020, Kavvas et al. [92] proposed the integration of these two methods to obtain results to a higher resolution, i.e., not just to analyze gene presence–absence, but also to analyze the links of allele variations to observed phenotypes, such as antimicrobial resistance. Kavvas’ group worked with *Mycobacterium tuberculosis* (TB) and built a genetic variant matrix (G) containing the information of all drug-tested TB strains and all their allelic variants. The abundance of AMR genes in the latest TB model (iEK1011) [93] justified the use of this model for their analysis. The Metabolic Allele Classifier (MAC), as they named it, is an allele-parameterized form of FBA, which takes the following as constraints: (i) an antimicrobial-specific objective function, (ii) a steady-state space, and (iii) upper and lower bounds for each reaction flux, in function to the information in G. The last two points describe strain-specific separate polytopes for resistance and susceptible strains within the overall flux space, and the first point is defined by the objective function that best separates both polytopes, where the optimal solution is the orthogonal vector to the plane that divides them, which is calculated with ML instead of the linear program. The MAC yields a drug susceptibility binary status as a final output: “susceptible” or “resistance” phenotype to a particular drug. Kavvas et al. tested three different AM compounds: pyrazinamide, para-aminosalicylic acid, and isoniazid, and performed pathway enrichment analysis out of the 197 alleles obtained, resulting in the identification of processes that were known AM mechanisms. Overall, Kavvas’ group’s work shed light on the possibility of studying allele variations–phenotype relations in a determined strain using just one GSMM of the same bacterium. Another example of the integration of the two methods (GSMM + ML) was reported by Pearcy et al. [94]. These authors aimed to reveal and describe the systemic relationship between the genetic determinants of AMR and metabolic evolutionary adaptations in *E. coli*. Using a set of unique *E. coli* genomes to analyze the AMR phenotypic variability, they developed a ML-integrated k-mer and SNP approach: they scanned entire genomes against selected phenotypes, allowing for the identification of arbitrary numbers of genomic features ranked on the strength of the correlation with the AMR and the susceptible phenotype (named the Gradient Boosting Classifier). These genetic determinants, such as AMR genes and specific mutations and alleles to specific phenotypes, were assessed using the GSMM: they used FBA to constrain the model and predict the effects (different metabotypes, see Figure 1) of the genetic determinants previously identified by performing gene deletion analysis in the metabolic network. Pearcy’s group identified 20 genes essential for growth under rich environment conditions, and the ones that had the highest importance depicted by the ML method may be promising dual-mechanism antimicrobial targets: they would inhibit essential metabolic pathways while reducing the bacterium’s ability to adapt. 

A problem that is sometimes difficult to address is the impact of the combinations of different drugs and/or how the drug efficacy (or the efficacy of drug combinations) changes depending on the growth conditions. To address this issue, Chung et al. [95] proposed an approach named Condition-specific Antibiotic Regimen Assessment using Mechanistic Learning (CARAMeL). This method simulates context-specific metabolic fluxes using GSMMs, constrained with environmental conditions, -omics data, and metabolite composition from control and mono- or multi-treatments. Obtained (simulated) fluxes are then used to train a ML model to predict interaction outcomes for novel drug combinations (using the Random Forests algorithm). Using *E. coli*’s and *M. tuberculosis*’ GSMMs, Chung’s group assessed their methodology using several scenarios with multiple drugs and clinical trials, inferring a set of differentially regulated genes by calculating the differential fitness (from chemogenomic data) and expression (from transcriptomic data). Then, they assigned four scores for each condition and trained ML models to associate the described profiles to drug combination outcomes, as interaction scores. After comparing the predictions with experimental data (-omics and literature) and assessing the performance for synergy and antagonism scenarios, the authors’ approach revealed to be accurate in predicting combination therapy outcomes. Afterwards, they assessed how a pathogen adapts to a first drug and how this, in turn, influences its sensitivity to a subsequent drug. Using data from evolved *E. coli* in single-drug treatments over different timespans and subsequent treatments, CARAMeL once again yielded robust predictions, also when addressing the model sensitivity to a broad range of drug combinations and growing media. 

Even when there have been other methodologies that do not use GSMMs [96], the joint effort of each of the approaches described in this review gives a broad perspective of the relationship between the metabolic state of bacteria and the effect of an antimicrobial. As described in the previous section, the nutrient contents of the bacterial medium can alter the metabolic state of bacteria, inducing different responses under AM stress, just as the results of 81,93] showed. Additionally, integrating different types of -omics data, GSMM, and ML, defining a context-specific objective function [92,97], and/or performing GE and ME analysis not only helps to understand the different AM metabotypes in bacteria, but also helps to shed light on the drug development and drug repurposing areas [71,74,78,79], in order to find antimicrobials able to kill bacteria with active metabolism, even if there is no growth. However, there are still some scenarios to address, such as the study of non-protein-coding regions, since it has been found that they can confer resistance as well [94,98].

Lastly, we would like to refer to a case study that used GSMMs predictions to enhance the killing activity of antibiotics. Brynildsen et al. [67,99] exploited the link existing between ROS production and antibiotic killing activity and used genome-scale metabolic modeling to identify metabolic targets that could lead to the overproduction of such ROS species and that could potentiate the action of several different antibiotics. Specifically, the metabolic network of *E. coli* was systematically perturbed (through the simulation of single-gene knockouts), and its flux distribution was analyzed to identify targets predicted to increase ROS production, i.e., genes whose removal from the WT would increase the intracellular ROS concentration. Remarkably, such computationally predicted targets were experimentally validated and demonstrated to confer increased susceptibility to oxidants and to killing by antibiotics. A schematic representation of the overall pipeline used in this work is reported in Figure 2B.

## 3. Future Directions

The use of genome-scale metabolic modeling has improved our understanding of the intertwined relationship between metabolism and antimicrobials. We believe that the most useful application of GSMMs resides in using them as scaffolds for the interpretation of -omics datasets, as this helps in reducing the space of possible solutions, for example, after an FBA simulation, and to identify the most likely set of active reactions in a specific condition. Indeed, the speed and ease-of-use of constraint-based simulations comes at the cost of many (sometimes unrealistic) constraints that must be imposed on the model to make it mathematically solvable. Among them, the one that most impacts the use of genome-scale metabolic modeling for studying the response of microbial cells to cellular perturbation is probably the identification of a proper objective function, i.e., the reaction in the metabolic network that describes the primary objective of the cell in a specific growth condition [100]. While it seems reasonable to assume that, in laboratory conditions (e.g., growth in a bioreactor), the generation of new biomass would be the main cellular objective of the population, we think that this prerequisite is hardly met in a stressful condition such as the exposure to antimicrobials. For this reason, FBA simulations that encompass the use of a biomass-oriented objective function to mimic an antimicrobial-exposed/tolerant metabotype might not be capable of adequately representing the pool of active reactions in the cell. While the majority of the studies (also reviewed in this work) have not directly tackled this issue up to now, others have observed that the exposure to antimicrobials and the evolution of resistance may introduce novel and still unexplored metabolic constraints and cellular objectives. This, in turn, hampers the identification of the actual objective function(s) that drive metabolic adaptation to antimicrobial exposure [23]. Computational approaches can be used to mitigate this effect, for example, by assuming that altered metabolite concentrations in antimicrobial-resistant strains may reflect an attempt to redirect intracellular fluxes toward specific but unknown metabolic objectives to drive and compensate for resistance [23,101]. In [92], the problem of correctly capturing the cellular objective of the antimicrobial-resistant phenotypes was circumvented by integrating GWAS information with the *E. coli* metabolic model and deriving an “antimicrobial-specific objective” function that provides novel insights regarding the metabolic basis of the mutations involved in antimicrobial resistance. This was achieved by determining the set of potential constraints imposed by an allele through discretization of the flux solution space; thus, deriving the metabotypes that more closely resemble the stressful conditions faced by cells. To tackle the same issue, i.e., the definition of a proper objective function under stressful conditions, Montezano et al. [97] studied the response of the *M. tuberculosis* bacterium when exposed to mefloquine, and integrated condition-specific proteomic data with its GSMM model to derive an antimicrobial-informed biological objective function, i.e., the actual cellular overall objective when facing antimicrobial-induced stress.

When experimental information on the cellular state in the presence of antimicrobials are not available, the exploration of the possible solution space through sampling methods (e.g., random flux sampling [101,102,103,104]) allows to define the possible set of active reactions without the necessity of defying a biomass-oriented objective function. In particular, random flux sampling can be used to characterize the solution space within a GSMM, allowing the identification of a statistically significant number of solutions that have been uniformly distributed throughout the entire solution space. In this way, an estimated probability distribution for each reaction’s flux in the network can be obtained without necessarily assuming growth as the primary objective function. Accordingly, random flux sampling might be a more reliable approach to simulate the metabolism of bacteria that are exposed to challenging conditions, such as the presence of antimicrobial molecules, in which the survival and the maintenance of basic metabolic functions with minimal cost may be more realistic objectives.

Population-level metabolic heterogeneity (i.e., cell-to-cell variation in metabolism within an isogenic population) is known to play a key role in the response of microbes to antimicrobial stress [105], and several pathogens have been shown to take on a distinct metabolic state in vivo to tolerate antimicrobials. Canonical genome-scale metabolic modeling approaches typically represent the averaged metabolic state of entire populations and, as such, are not suited to capture single-cell metabolic differences. Additionally, this within-population variability seems to have a stochastic basis [106], another feature whose modeling with a constraint-based approach has been seldomly attempted in the past. Recently, however, computational methods to tackle both the metabolic heterogeneity within a population and the stochastic nature of certain cellular processes and their impact on metabolism have been developed [107,108]. Besides revealing important basic insights into the possible consequences of the stochasticity that is implicit in some biological circuits, these methods pave the way for their exploitation in the study of the metabolic state of bacterial cells in a population in the presence of antimicrobials. Since, for example, persisters are known for their lowered metabolic state, such an approach could indicate which pathways should be revived to reduce the fraction of dormant persister cells, and thus to develop metabolic-oriented and more effective therapies. Indeed, there could be different and independent ways through which metabolic dormancy is reached (i.e., downregulation of different metabolic pathways) within a population, and genome-scale metabolic modeling might help in identifying which of them is/are actually playing a role in persisters’ survival. In this context, methods such as BacDrop [109] that permit to elucidate the heterogeneous responses of microbial cells to antimicrobial stress using single-cell RNA-sequencing on millions of single bacterial cells could permit adding a further layer of information (constraints) to the model, thus refining the predictions. Overall, there seems to be a methodological gap in the repertoire of computational approaches able to simulate the (sub-optimal) metabolic landscape of persister cells. Additional effort is thus required to develop tolls and theoretical frameworks to generate testable hypotheses on the mechanisms through which tolerance/persistence is achieved at the whole population level.

As we saw in the case studies, the modeling of antimicrobial-induced metabolic changes has recently been boosted by the combination of ML methods with GSMMs. There are (at least) three different ways for integrating ML with constraint-based metabolic modeling (see Sahu et al. [110] for a thorough review), and ML can be used to identify those cellular components (e.g., genes, transcripts, metabolites) that should actually be included in the corresponding metabolic model [111,112]. Other ML methods can be directly integrated with the FBA algorithm [113]. Finally, the outcomes (flux distributions) of an FBA simulation can be analyzed and further refined by using ML. In the context of antimicrobial stress, a flux-based ML model was recently used to simulate the impact of metabolic heterogeneity on drug interactions and to predict the outcomes of novel and unseen combinations of drugs in *E. coli* and *M. tubercolosis*. It is important to stress that the relevance of such hybrid models (and of GSMMs in general) resides in their capability of providing a mechanistic understanding of how biological circuits (metabolism in this case) respond to environmental stresses/fluctuations, and which strategies are the most promising in preventing antimicrobial resistance/tolerance/persistence [114].

## Figures and Tables

**Figure 1 antibiotics-12-00896-f001:**
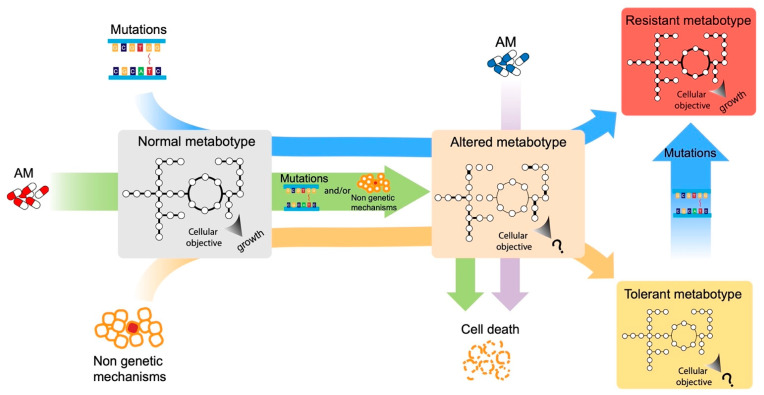
Possible paths to resistant and tolerant metabotypes. Here, we show the hypothetical effects of mutations, antimicrobials’ exposure, and non-genetic mechanisms on microbial metabotypes, including the effects on the single reactions (thicker links among the nodes) and the most likely objective function (“growth” or “?”).

**Figure 2 antibiotics-12-00896-f002:**
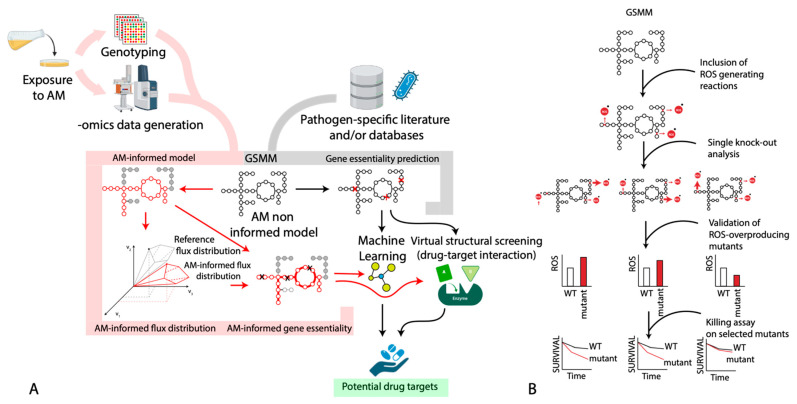
Schematic representation of AM-informed and AM-non-informed GSMM. (**A**) The combination of automatic metabolic reconstruction tools, literature, and strain-specific experiments leads to the creation of general, strain-specific GSMM models that can be further used for downstream simulations, leading, for example, to the prediction of possible drug targets. The integration with -omics data obtained from ad hoc designed experiments permits the generation of context-specific GSMM, capable of mimicking in a more reliable way the metabolic state of cells in the tested experimental conditions. Such AM-informed models can then be used to study the differences in flux distributions under such stressful conditions and/or to obtain to a more focused identification of condition-specific gene essentiality patterns. (**B**) The approach proposed by Brynildsen et al. [67], to demonstrate the link between ROS production and antibiotic efficacy. Including ROS-producing reactions in the original *Escherichia coli* GSMM and using simulations of gene knockouts, they identified targets (genes) whose removal would lead to an increased intracellular production of ROS. These mutant strains were then (successfully) tested for increased ROS production and increased sensitivity to antibiotics’ killing.

## Data Availability

No new data were created or analyzed in this study. Data sharing is not applicable to this article.

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
