# Peer review of "Understanding Antimicrobial Resistance Using Genome-Scale Metabolic Modeling"

_antibiotics, 2023, doi:10.3390/antibiotics12050896_

Round 1
Author Response
R1
In this review paper “Genome-scale metabolic modeling and antimicrobial resistance” Alonso-Vásquez et al. have discussed the antimicrobial resistance (AMR), how the microorganisms remodel their metabolism and become resistant to antimicrobial agents posing a huge challenge threatening global public health. The authors have given more stress on cellular process especially the metabolism and how antimicrobials can reprogram the metabolic phenotype of microbial cells that may lead to altered metabotypes. As the genome scale metabolic models can link metabolic states with phenotypes, we can use GSMM integrated with context or condition specific HT data to understand the microbial metabolic reprogramming in response to antimicrobial exposure.
Throughout the article, the author has used too many commas, statements in () and long sentences making it difficult to follow and understand.
I am recommending this review for major revision.
See my comments for each section:
Major comments
- Title:
Title of the manuscript should reflect broader and applied aspects that have been discussed in the paper, e.g., “Applications of genome-scale metabolic modeling to understand the antimicrobial resistance” or sometwhat like that.
R: We thank the reviewer for this comment, and we modified the title in “Understanding antimicrobial resistance using genome-scale metabolic”
- Abstract:
The wording of the abstract should be changed to more scientific language.
Words or phrases like:
line 8, 18: tackle,
line 14: hurdles,
line 15: external conditions,
line 16: For this reason,
line 19: topic.
R: We have replaced the words suggested by the reviewer with synonyms trying to maintain the sense of the abstract. In case it doesn't work we can change it again, but we would appreciate more specific comments on how to implement the changes.
line 8, 18: tackle: to fight
line 14: hurdles: problems
line 15: external conditions: the environment
line 16: For this reason. To solve this problem
line 19: topic: We removed this topic and the brackets.
The whole sentence (line 18-20) needs to be re written to make it clear and easy to understand.
More information about how GSMM can be used to understand the cellular response in understanding the antimicrobial resistance and how this knowledge gap can be addressed using metabolic modeling should be made punching line of the abstract.
Overall, there is scope to improve the abstract.
R: We thank the reviewer for this comment. However, the journal has a maximum limit of 200 words for the abstract so we cannot add further explanations in the abstract.
- Introduction:
Line 26: of our century the/this century
- We modified the text accordingly.
Line 29: Given this threat make appropriate changes.
R: We have replaced “Given this threat” with “Consequently”.
Line 30: wide variety of change the word variety.
R: We have replaced “a wide variety of” with “Several”.
Line 31-34: Sentence is too long.
R: We have split the sentence into two separate sentences.
Line 35-36: Are these few specific metabolic genes or from all the microbial genes?
R: In the text we do not name metabolic genes, so we're talking about bacterial genes in general.
Line 36-38: too many comas. Try to rephrase the sentence.
R: We rephrase this sentence:
“However, this description represents only the immediate effects of the antimicrobial. In many cases what is not known and is still the subject of debate, is how antimicrobials actually kill bacterial cells.“
Line 38: Certainly … does not fit the context.
R: We removed the word “Certainly” from the text.
Line 40-48: The author is trying to combine too many concepts into one or two sentences which
is making it very complex to understand the flow.
R: We have tried to simplify this paragraph.
Line 49-53: Overall, antimicrobials can affect bacterial metabolism in three ways. First of all, they have a direct effect on metabolism that affect their efficacy …
The second and third scenario has not been motioned throughout the introduction.
R: The second and the third ways are already present in the text, however, for further clarity, we have better specified in the text what they are.
Line 71-76: As mentioned before, however, also ….
The sentence is too lengthy and complex. Try to rephrase.
R: We have split the sentence into three separate sentences.
Line 78-83: However, antimicrobials can influence bacterial metabolism also in two indirect ways … Which are those two indirect ways? Although, the author may have mentioned it but I did not see it as way one and way 2.
R: The two ways are already present in the text, however, for further clarity, we have better specified in the text what they are.
Lines 93-110: Statements are too vague and do not fit with the flow.
R: This paragraph is necessary to introduce the definition of the terms: resistance, heteroresistance, tolerance and persistence, that are the words used to describe the ways in which a bacterial cell can react to an antibiotic. Since the metabolic state of bacterial cells could affect these responses, we need to introduce in the text these definitions, in order to allow the reader to understand the following paragraphs.
Lines 178: The introduction to GSMMs should start with new paragraph.
R: Done as suggested.
Lines 180-210: The GSMM part can be explained in more detail.
R: Done as suggested.
Tabulate the list all the microbial metabolic models that have been used in anti-microbial resistance studies that includes organism name and list of their target genes or metabolic pathways.
R: In this new version of the manuscript, we have included a supplementary table (Table S1) listing all the metabolic reconstructions that we cite and that have been used in anti-microbial resistance studies.
Figure 1: There is no novelty in the figure.
R: The purpose of this figure is to schematically illustrate the concepts that are presented along the text. Being this manuscript a survey of previously developed methods and study-cases, it is hard to introduce some novelty in the figure and/or in the text (except for highlighting future perspectives and/or knowledge gaps as we did in the final part of the work). Anyways, we are ready to take into consideration more specific comments from this reviewer on how to introduce some novelties in this figure.
- Case studies
Lines 217-558: Mention and Include cellular objective functions used with each case, target pathways, methods used for multi-omics data integration to create context or condition specific models. Demonstrate a case study example from published models with simulation results and how cellular metabolism was reprogrammed with application to microbial resistance. Include a figure from the example study.
R: We have included the objective functions used in each study in the new supplementary table that we have produced for this revised version of the work (Table S1). As for the case study, we have added the following work to the list of the examples [1]. In this work, the authors observed that, in most cases, it is the antibiotics-induced elevated concentration of ROS species that kills the bacteria. For this reason, they used genome scale metabolic modelling to identify metabolic targets that could lead to the overproduction of such ROS species and that could potentiate the action of several different antibiotics. Specifically, the metabolic network of E. coli was systematically perturbed and its flux distribution analyzed to identify targets predicted to increase ROS production. Remarkably, such computationally predicted targets were validated experimentally and demonstrated to confer increased susceptibility to oxidants and to killing by antibiotics. We have added a new panel to Figure 2 that sums up the strategy used in this work.
Figure 2: there is no novelty in the figure. It is not clear what role ML does play?
R: The purpose of this figure is to schematically illustrate the concepts that are presented along the text. Being this manuscript a survey of previously developed methods and study-cases it is hard to introduce some novelty in the figure and/or in the text (except for highlighting future perspectives and/or knowledge gaps as we did in the final part of the work). Anyway, we are ready to take into consideration more specific comments from this reviewer on how to introduce some novelties in this figure. As for the role of ML in GSMM, ML is typically used to broaden the space of predictions, combining the outcomes of FBA simulations (flux distributions) with other data types (as explained in the text). We have slightly modified Figure 2 to make it clearer the role of ML in the overall pipeline.
- Future directions
Lines: 562-650:
The authors should also include the GSMM applications in identification of potential drug targets and explain in detail how machine learning and pharmacokinetic/pharmacodynamic modeling can be integrated with metabolic modeling. This will provide overall mechanistic insights and treatment plans for antimicrobial-resistant diseases. The overall future directions should be focused on applications of GSMMs in antimicrobial pharmacology.
R: Done as suggested. This is described in two of the case studies: the one from Kavvas et al. [2] and the one from Pearcy et al. [3]. These examples have been included before the “Future directions” section.
Cited Bibliography
- Brynildsen, M.P.; Winkler, J.A.; Spina, C.S.; MacDonald, I.C.; Collins, J.J. Potentiating Antibacterial Activity by Predictably Enhancing Endogenous Microbial ROS Production. Nat. Biotechnol. 2013, 31, 160–165, doi:10.1038/nbt.2458.
- Kavvas, E.S.; Yang, L.; Monk, J.M.; Heckmann, D.; Palsson, B.O. A Biochemically-Interpretable Machine Learning Classifier for Microbial GWAS. Nat. Commun. 2020, 11, doi:10.1038/s41467-020-16310-9.
- Pearcy, N.; Hu, Y.; Baker, M.; Maciel-Guerra, A.; Xue, N.; Wang, W.; Kaler, J.; Peng, Z.; Li, F.; Dottorini, T. Genome-Scale Metabolic Models and Machine Learning Reveal Genetic Determinants of Antibiotic Resistance in Escherichia Coli and Unravel the Underlying Metabolic Adaptation Mechanisms. mSystems 6, e00913-20, doi:10.1128/mSystems.00913-20.
Reviewer 2 Report
The work presents computational approaches in the field of genome-scale modelling (and connected fields) for the study of microbial response to antimicrobial exposure. The introduction describes the biological background including the various roles of metabolism in antimicrobial resistance and the distinction between antimicrobial resistance, tolerance, and persistence. The main body of the article presents recent works broadly dividing them based on the application of experimentally contenxtualised genome-scale metabolic models (GSMMs) and non-contextualised GSMMs, covering also emerging trends such as their combination with other methodologies. Discussed future directions include the development of more meaningful objective functions (or alternatively the use of objective-independent methods), the exploration of population heterogeneity, and the integration of GSMMs and machine learning.
The addressed topic is of high relevance to the field and timely summarises an important area of research. The work is clearly written and can be relevant both to computational biologists and non specialists, with an appropriate balance between technical and applied aspects. The references are in adequate number and scope, without excessive self-citation, while the figures clearly convey the key points. The use of language is sound and only a few typos are present.
The main weakness in my opinion is a relative lack of cohesion that can create confusion and reduce the effectiveness of the work. There are a few aspects that therefore deserve attention by the authors.
- A comparison with other recent surveys is missing. While this work is not fully comparable with others in terms of scope, I believe it has some overlap with some of them and deserves a brief contextualisation: 10.1038/s41429-020-00366-2, 10.1016/j.engmic.2022.100021, 10.3389/fcell.2020.566702. These (and in principle others as well) could be probably mentioned at the end of the introduction.
- While the introduction nicely introduces many biological aspects around antimicrobial action, the modelling counterparts for some of these aspects is not described/discussed in detail in the rest of the article. I understand that in some cases this might be due to a lack of relevant literature on a specific aspect, but even in such cases potential research gaps could be highlighted. For example, the study of the metabolic mechanisms of action of bacteriostatic antimicrobials, as opposed to bactericidals, is an interesting distinction that could be highlighted more on the computational side. The same goes for the resistance/tolerance/persistence mechanisms. For instance, are there any specific approaches/challenges for modelling any of these biological phenomena? Has any mechanism been explicitly modelled (e.g. target modification, drug inactivation, drug transport) other than through omics data integration and in silico gene knock-out (e.g. stoichiometrically)? While there are many potential aspects to be discussed and not all of them need to be, implementing some of them would better connect the introduction with the rest of the paper.
- In my view, it would make sense to present the non-contextualised GSMM approaches first and the contextualised ones second, as the latter build on the former. For example, gene essentiality analysis is first described at the beginning of Section 2.1 yet mentioned several times in 2.2. This is mainly a conceptual point though, and if the authors think that the current order better serves the conceptual flow then it is fine.
- Some studies described in Section 2.2 (non-contextualised GSMM approaches) include the use of contextualised GSMMs (e.g. 77 and 79), which can be confusing. If the same studies employed both contextualised and non-contextualised approaches, I believe that each aspect should be separately described in the corresponding section.
- In the main body of the article (Section 2), the organisation of the content is at times confusing and makes it difficult to follow some individual sub-topics while also creating some redundancy. As the section is divided in only two broad subsections, I think it is important to pay attention to the conceptual flow across paragraphs to reduce the risk for a reader of getting lost. For example, the reuse of previously curated GSMMs is described in the paragraph starting at line 284, but the same sub-topic is already touched on at lines 266-275.
Specific comments:
Page 8, line 387. It should be Section 2.2.
Page 9, line 439. This sentence is not clear and seems incomplete.
Page 13, line 598. It may not be clear how the mentioned “antimicrobial-informed biological objective function” could be defined. More details would in general be helpful to better understand this approach.
Figure 1. The figure would be more clear with the text of a more homogeneous size. The text inside the boxes and the arrows is slightly too small to be read without zooming in.
Author Response
R2
The work presents computational approaches in the field of genome-scale modelling (and connected fields) for the study of microbial response to antimicrobial exposure. The introduction describes the biological background including the various roles of metabolism in antimicrobial resistance and the distinction between antimicrobial resistance, tolerance, and persistence. The main body of the article presents recent works broadly dividing them based on the application of experimentally contenxtualised genome-scale metabolic models (GSMMs) and non-contextualised GSMMs, covering also emerging trends such as their combination with other methodologies. Discussed future directions include the development of more meaningful objective functions (or alternatively the use of objective-independent methods), the exploration of population heterogeneity, and the integration of GSMMs and machine learning.
The addressed topic is of high relevance to the field and timely summarises an important area of research. The work is clearly written and can be relevant both to computational biologists and non specialists, with an appropriate balance between technical and applied aspects. The references are in adequate number and scope, without excessive self-citation, while the figures clearly convey the key points. The use of language is sound and only a few typos are present.
The main weakness in my opinion is a relative lack of cohesion that can create confusion and reduce the effectiveness of the work. There are a few aspects that therefore deserve attention by the authors. - A comparison with other recent surveys is missing. While this work is not fully comparable with others in terms of scope, I believe it has some overlap with some of them and deserves a brief contextualisation: 10.1038/s41429-020-00366-2, 10.1016/j.engmic.2022.100021, 10.3389/fcell.2020.566702. These (and in principle others as well) could be probably mentioned at the end of the introduction.
R: This is a very good suggestion and the reviews were mentioned at the end of the introduction.
- While the introduction nicely introduces many biological aspects around antimicrobial action, the modelling counterparts for some of these aspects is not described/discussed in detail in the rest of the article. I understand that in some cases this might be due to a lack of relevant literature on a specific aspect, but even in such cases potential research gaps could be highlighted. For example, the study of the metabolic mechanisms of action of bacteriostatic antimicrobials, as opposed to bactericidals, is an interesting distinction that could be highlighted more on the computational side. The same goes for the resistance/tolerance/persistence mechanisms. For instance, are there any specific approaches/challenges for modelling any of these biological phenomena? Has any mechanism been explicitly modelled (e.g. target modification, drug inactivation, drug transport) other than through omics data integration and in silico gene knock-out (e.g. stoichiometrically)? While there are many potential aspects to be discussed and not all of them need to be, implementing some of them would better connect the introduction with the rest of the paper.
R: We thank the reviewer for highlighting this important aspect of the work. In our opinion, the points raised here ultimately connect to two main aspects: i) the case of bactericidal vs. bacteriostatic effects of antimicrobial drugs overall relates to the actual cellular objective function when facing the compound and on the effects of bacteriostatic drugs on metabolism (detailed explained in [1]). ii) The case of resistance/tolerance/persistence relates both to the cellular objective function and to the intrinsic stochasticity of microbial populations, in that persisters seem to represent stochastically formed sub-populations with “repressed metabolism”. We agree with the reviewer that there exists a lack of relevant literature in the context of these two topics. This is the reason why, in our work, we discussed these two topics in the “Future perspectives” section, highlighting the need for implementing new computational strategies to account for “alternative” objective functions (e.g. those exploited by cells facing bacteriostatic drugs) and population-level heterogeneity (likely a key determinant, together with metabolism downregulation, of persistence). In this revised version of the work, we have included a few more examples that have investigated such features (e.g. the modelling of ROS production in the context of bactericidal and bacteriostatic activity of some drugs [2]) and try to better connect these (namely persistence and bactericidal vs. bacteriostatic actions) with possible computational pipelines to account for them. Also, we have accounted for the lack of computational approaches to simulate the (sub-optimal) metabolic landscape of persister cells (see the revised version of the “Future directions” paragraph).
- In my view, it would make sense to present the non-contextualised GSMM approaches first and the contextualised ones second, as the latter builds on the former. For example, gene essentiality analysis is first described at the beginning of Section 2.1 yet mentioned several times in 2.2. This is mainly a conceptual point though, and if the authors think that the current order better serves the conceptual flow then it is fine.
R: We appreciate the suggestion and did the pertinent changes. This particular change has been made in the text without using the “track changes” function, to not generate too much confusion in the text.
- Some studies described in Section 2.2 (non-contextualised GSMM approaches) include the use of contextualised GSMMs (e.g. 77 and 79), which can be confusing. If the same studies employed both contextualised and non-contextualised approaches, I believe that each aspect should be separately described in the corresponding section.
R: Certainly, the concept of contextualized models could create some confusion. Nevertheless, here we specify contextualized with data retrieved from antimicrobial treatments only (AM-informed and AM-non informed GSMM). We made the pertinent changes to the manuscript avoid any misunderstanding.
- In the main body of the article (Section 2), the organisation of the content is at times confusing and makes it difficult to follow some individual sub-topics while also creating some redundancy. As the section is divided in only two broad subsections, I think it is important to pay attention to the conceptual flow across paragraphs to reduce the risk for a reader of getting lost. For example, the reuse of previously curated GSMMs is described in the paragraph starting at line 284, but the same sub-topic is already touched on at lines 266-275.
R: We agree on the complexity of our text, especially in section 2. It is to be said that it is quite hard to unambiguously classify all the works that we cite in the text as many of them actually fall in more than one category. We think that the order in which we are currently presenting the works we cite is surely only one of the many possible but is somehow consistent throughout the manuscript and, for this reason, we would like to keep it in its current form. We are open, however, to take into consideration specific rearrangements that the reviewer would suggest. As for the case cited (line 284 of the older version of the text) we rephrased the text to make it clear that we considering an aspect that has been already encountered a few lines above.
Specific comments:
Page 8, line 387. It should be Section 2.2.
R: Corrected.
Page 9, line 439. This sentence is not clear and seems incomplete.
R: Corrected.
Page 13, line 598. It may not be clear how the mentioned “antimicrobial-informed biological objective function” could be defined. More details would in general be helpful to better understand this approach.
R: We appreciate the suggestion and did try to describe how the antimicrobial-informed biological objective function (or antimicrobial-specific objective function) is calculated in the space of solutions. However, it would require a deep mathematical explanation to describe every step, which could be found in the paper [3]. We have added a short explanation on what we mean with “antimicrobial-informed biological objective function”.
Figure 1. The figure would be more clear with the text of a more homogeneous size. The text inside the boxes and the arrows is slightly too small to be read without zooming in.
R: We have made the font outside of the boxes homogeneous (28 pts) and increased the text inside the boxes as much as possible.
Cited Bibliography
- Lobritz, M.A.; Belenky, P.; Porter, C.B.M.; Gutierrez, A.; Yang, J.H.; Schwarz, E.G.; Dwyer, D.J.; Khalil, A.S.; Collins, J.J. Antibiotic Efficacy Is Linked to Bacterial Cellular Respiration. Proc. Natl. Acad. Sci. U. S. A. 2015, 112, 8173–8180, doi:10.1073/pnas.1509743112.
- Brynildsen, M.P.; Winkler, J.A.; Spina, C.S.; MacDonald, I.C.; Collins, J.J. Potentiating Antibacterial Activity by Predictably Enhancing Endogenous Microbial ROS Production. Nat. Biotechnol. 2013, 31, 160–165, doi:10.1038/nbt.2458.
- Kavvas, E.S.; Yang, L.; Monk, J.M.; Heckmann, D.; Palsson, B.O. A Biochemically-Interpretable Machine Learning Classifier for Microbial GWAS. Nat. Commun. 2020, 11, doi:10.1038/s41467-020-16310-9.
Reviewer 3 Report
In this review article, Perrin and coworkers explored the new targets and strategies to tackle antimicrobial resistance. They investigated into several modeling approaches for genome-scale metabolic engineering and the ease at which a genome sequence can be converted into models to run basic phenotypes predictions. Moreover, they demonstrated the use of computational modeling to study the relationship between microbial metabolism and antimicrobials and reviewed the recent advances in application of genome-scale metabolism modeling to the study of microbial response to antimicrobial exposure. In summary, this review article was comprehensive, well-written, and covered a wide range of topics that have correlations to genome-scale metabolic modeling and antimicrobial resistance. Only some minor changes need to be addressed before final publication.
Comments:
1) In line 183-184, the authors mentioned the resulting models were validated using several known databases such as KEGG, PubMed, and BiGG. Is there a preference of which database to use? It would be better if the authors could demonstrate more in details about the advantages and disadvantages of these different platforms.
2) Could the genome-scale metabolic modeling be used to analysis natural products and biosynthetic gene clusters (BGCs)? Because a lot of BGCs analysis require huge amount of data, so I am wondering whether this modeling work would help with the BGCs selection?
3) The authors mentioned several groups have proposed the combination of GSMM and machine learning in order to broad the space of predictions. Have these machine learning models been validated? And where were the original data coming from to train the machine learning model? Is artificial intelligence also can be involved in the “training model” portion? And are there any automation strategies being used to deal with these big sets of data?
4) In line 460, I am wondering how was the biomass flux being calculated? What formulation does it require?
5) It seems there will be tons of omics data being generated by GSMM. How can people analyze such big datasets? I think besides the current solutions, the authors can also think about some high-throughput screening strategies to deal with data processing and auto-analysis.
6) For Figure 2, the icons in the “virtual structural screening” part need to be explained in detail. Currently, it is hard to understand what the icons represent for. And the text inside the icons is too small to read.
Author Response
R3
In this review article, Perrin and coworkers explored the new targets and strategies to tackle antimicrobial resistance. They investigated into several modeling approaches for genome-scale metabolic engineering and the ease at which a genome sequence can be converted into models to run basic phenotypes predictions. Moreover, they demonstrated the use of computational modeling to study the relationship between microbial metabolism and antimicrobials and reviewed the recent advances in application of genome-scale metabolism modeling to the study of microbial response to antimicrobial exposure. In summary, this review article was comprehensive, well-written, and covered a wide range of topics that have correlations to genome-scale metabolic modeling and antimicrobial resistance. Only some minor changes need to be addressed before final publication.
Comments:
- In line 183-184, the authors mentioned the resulting models were validated using several knowndatabases such as KEGG, PubMed, and BiGG. Is there a preference of which database to use? It would be better if the authors could demonstrate more in detail about the advantages and disadvantages of these different platforms.
R: This is a good question. The quality tends to be higher if there is a wide validation across several databases.
- Could the genome-scale metabolic modeling be used to analysis natural products and biosynthetic gene clusters (BGCs)? Because a lot of BGCs analyses require huge amounts of data, so I am wondering whether this modeling work would help with the BGCs selection?
R: Yes. Indeed, GSMMs have been originally developed for the identification of the possible targets for improving the biosynthesis of biotechnologically valuable compounds. To make this point clearer, we have modified the Introduction adding a few references.
- The authors mentioned several groups have proposed the combination of GSMM and machine learning in order to broaden the space of predictions. Have these machine learning models been validated? And where were the original data coming from to train the machine learning model? Can artificial intelligence also be involved in the “training model” portion? And are there any automation strategies being used to deal with these big sets of data?
R: The machine learning models are trained with the data obtained from simulations using GSMM. These simulations are performed using GSMMs constrained with environmental conditions, -omics data and metabolite composition from control and mono or multi-treatments. Since machine learning is a method to achieve artificial intelligence, AI is indeed involved in the training model process, being as well one of the best strategies so far to deal with these big sets of data.
- In line 460, I am wondering how was the biomass flux being calculated? What formulation does it require?
R: The biomass flux value was previously obtained using Flux Balance Analysis, and afterwards, the authors used the 90% of that simulated value to fix the biomass objective function boundaries of the Flux Sampling analysis.
- It seems there will be tons of omics data being generated by GSMM. How can people analyze such big datasets? I think besides the current solutions, the authors can also think about some high-throughput screening strategies to deal with data processing and auto-analysis.
R: We thank the referee for this good question. In general, -omics data are obtained from in vitro/in vivo experiments, i.e. retrieved directly from the organism to study. This means to obtain the abundance of multiple mRNA transcripts for transcriptomic data, the high-throughput characterization of the metabolites of the cell for metabolomic data, the identification and quantification of the complete set of proteins for proteomic data, to name a few. Once these data have been obtained, they can be integrated with GSMM in order to be analyzed. This part of GSMM analysis falls in the classification that, in the manuscript, we have named “AM-informed models”
- For Figure 2, the icons in the “virtual structural screening” part need to be explained in detail. Currently, it is hard to understand what the icons represent for. And the text inside the icons is too small to read.
R: Figure 2 has been modified also according to Reviewer 2 suggestions. We have increased the size of the text in the figure and the size of the icon. As for the Virtual Structural screening” part, we have added that the “virtual structural screening” refers to the prediction of the drug-target (physical) interaction.
Round 2
Reviewer 1 Report
The title: "Understanding antimicrobial resistance using genome-scale metabolic" is incomplete. The authors have not included the modeling term. The title can be modified accordingly to "Understanding antimicrobial resistance using genome-scale metabolic modeling"
Reviewer 2 Report
I am glad that the authors found the points raised useful, which were all addressed in full.